# Evaluation method for cell-free in situ tissue-engineered vasculature monitoring: Proof of growth and development in a canine IVC model

**Goki Matsumura**[1,2]*, **Noriko Isayama**[2,3], **Hideki Sato**[4]

**1** Medical Safety Management Department of Tokyo Women's Medical University, Shinjuku-ku, Tokyo, Japan, **2** Department of Cardiovascular Surgery, The Heart Institute of Japan, Tokyo Women's Medical University, Shinjuku-ku, Tokyo, Japan, **3** Ueno no Mori Animal Clinic, Tokyo, Japan, **4** Gunze Ltd., QOL Research Center, Kyoto, Japan

* matsumura.goki@twmu.ac.jp

**Data Availability Statement:** All relevant data are within the manuscript and its Supporting Information files.

## Abstract

We previously developed a non-cell-dependent biodegradable scaffold to create in situ tissue-engineered vasculature (iTEV) and tested it in a canine inferior vena cava (IVC) model. As iTEV features change dramatically during tissue generation, practical, simple, and accurate methods to evaluate iTEV are needed. The present study examined the usefulness of a novel method to evaluate iTEV growth and remodeling according to a simple formula using angiography: hepatic vein (HV) index = (IVC–HV junction angle) ÷ (π × [minimal internal iTEV diameter ÷ 2]$^2$). HV index strongly correlated with the pressure gradient across iTEV, which tended to improve during the tissue generation period up to 12 months post-implantation. Time-course changes in HV index reflected iTEV tissue development and in-vivo characteristics, such as hemodynamic congestion. In conclusion, HV index is useful to assess iTEV graft function because it represents both the morphometrics and hemodynamics of iTEV with only diagnostic imaging data.

## Introduction

We previously reported the usefulness of a tissue-engineered graft (TEG) created with a biodegradable scaffold to accomplish vascular continuity in animal models [1–5] and clinical applications [6–9]. Implanted biodegradable materials act as templates for invasive cells from surrounding tissues, seeded cells, and/or newly generated tissues. However, with simultaneous changes in material degradation and tissue development, TEG features dramatically change as vascular tissue matures into native tissue [2].

In terms of the features of a developing TEG, our research focuses on the development of optimal materials for appropriate cell proliferation. These materials possess suitable repair characteristics to address the decrease in the internal diameter of TEGs during tissue generation. However, concerns of narrowing and stenosis during early tissue development of TEGs

**Funding:** This work was supported in part by KAKENHI (Grant-in-aid for Scientific Research) of the Japan Society for the Promotion of Science (JSPS) (C: number 17K10739, C: number 23591878) from the Ministry of Education, Culture, Sports, Science and Technology (MEXT), Japan, and in part by an Open Research Grant from the Japan Research Promotion Society for Cardiovascular Diseases (2018 and 2019). The funders had no role in study design, data collection, analysis, decision to publish, or preparation of the manuscript.

**Competing interests:** The authors have declared that no competing interests exist.

are still prevalent, even with tissue-engineering strategies that use better biodegradable constructs.

In-vivo tissue regeneration of TEGs is dependent on each individual's self-remodeling/ regeneration ability. These individualized variations equate to a range of pressure gradients across the TEG, which influence TEG narrowing and may cause non-crucial, but congestive, changes in the liver and other organs during tissue regeneration. As previously reported, changes in immature TEGs improve as the tissue generates and remodels into native-like tissue [3, 5, 10]. If there are pressure gradients across the TEG with congestive changes during this tissue generation period, the distal side of the suture line (namely the inferior vena cava [IVC] and hepatic vein [HV]) distends or dilates due to pressure overload. These unwanted changes were observed in our previous animal experiments. This negative characteristic prompted us to establish a novel evaluation method for TEG assessment. The idea was based on the notion that if we can identify the relationship among pressure gradient, TEG diameter, and IVC–HV junction angle, we can speculate the pressure gradient and evaluate the degree of liver congestion using imaging observations alone. Furthermore, we can compare changes in TEG features over time and calculate the degree of vascular remodeling using a logically calculated value. The present study was thus performed to assess the relationship among pressure gradient, TEG diameter, and IVC–HV junction angle in an animal model of TEG. Our findings may be applicable in a clinical setting.

In this study, a canine IVC model with in situ tissue-engineered vasculature (iTEV) was used because it does not require any cells, medication, or materials other than a tubular biodegradable scaffold to create a self-regenerated vasculature. As mentioned above, iTEV increases in wall thickness due to intimal hyperplasia, followed by healing and remodeling of related tissues to within a normal range; thus, narrowing or stenotic changes in iTEV are unavoidable [2, 5, 10]. The pressure gradient across iTEV increases, while the internal diameter decreases; however, this improves with time, allowing iTEV to develop into a native-like vessel [5]. In terms of the features of iTEV with proper tissue regeneration, iTEV ultimately remodels into the shape required by the living organism, without an unacceptable pressure gradient. However, because iTEV relies on each individual's self-renewal ability, and each organism has its own healing/remodeling process, evaluating iTEV in each individual sample over time is difficult.

In the present study, iTEV was followed by catheterization to obtain angiographic imaging data and hemodynamic values. iTEV images were then analyzed to explore the usefulness of this novel evaluation method. We used a formula consisting of the smallest iTEV diameter and the IVC–HV junction angle. We named the novel score calculated with this simple formula the "HV index" to evaluate iTEV remodeling. The HV index enables congestive changes in iTEV to be monitored over time. An improvement in congestion suggests improvements in pressure gradient, remodeling, adaptation, and/or growth of iTEV, irrespective of body weight or height.

One of the issues with tissue-engineered vasculature is that the features of the vasculature dramatically change with time; thus, it is difficult to compare graft function by assessing the shape of iTEV itself. Furthermore, it is difficult to chronologically evaluate changes in iTEV using neither pressure gradient nor chronologically changing features alone because the pressure gradient and congestive changes do not recover simultaneously, and tissue generation patterns differ between individuals. Accordingly, the present study aimed to assess the usefulness of HV index for time-course evaluation of iTEV.

We also evaluated iTEV maturation using chronological development of vascular smooth muscle cell (VSMC) phenotypes to show that HV index yields robust results for the evaluation of iTEV from the perspective of biochemical analyses because VSMC maturity reflects iTEV

maturity [2, 5, 10]. Cells in developing iTEV include cells found in healthy tissues, inflammatory cells, and adhesive cells, as well as fibrous tissue, and each of these components increase or decrease in number with tissue development. Thus, we speculated that an internal control such as beta-actin, which involves these inflammatory cell proteins, is not appropriate to examine VSMCs in developing iTEV. Furthermore, the cell composition and number of cells differ between individuals. Accordingly, we evaluated chronological changes in VSMC differentiation by examining the ratio of desmin, vimentin, and myosin heavy chain 11 (MYH11) proteins to α-smooth muscle actin (SMA) to demonstrate time-dependent differentiation of iTEV and to show the development of iTEV from the perspective of biochemical assessment. Accordingly, iTEV development was also assessed by examining the increment of differentiated VSMCs within the iTEV.

Herein, we describe HV index as a novel evaluation method. This method considers both morphometric and hemodynamic metrics of iTEV for precise evaluation of iTEV growth and remodeling. With the help of biochemical and immunohistological analyses, iTEV growth and remodeling are illustrated.

## Materials and methods

### In-vivo experimentation

All animals received care according to the National Institutes of Health (NIH) guide for the care and use of laboratory animals (NIH Publications No. 8023, revised 1978). The ethics committee of Animal Research at Tokyo Women's Medical University reviewed and approved the study protocols (Permit Numbers: AE20-008-3-C, AE19-001, AE18-102, AE17-1, and AE16-1). All experiments were conducted at the animal experimentation facility of The Heart Institute of Japan.

In total, 26 healthy adult female beagles (Rikaken, Inc., Tokyo, Japan) with a mean weight of 9.4 kg (8.5–10.5 kg) were used in this study. Six animals were euthanized at 1, 3, and 6 months post-implantation of iTEV, and 8 animals were euthanized at 12 months for angiography, pressure studies, and histological and biochemical analyses.

For surgery, animals were anesthetized with intravenous injection of midazolam (0.1–0.3 mg/kg body weight); continuous intravenous injection of propofol (0.1–0.3 mg/kg/min); and intramuscular injection of medetomidine hydrochloride (5.0 μg/kg body weight), butorphanol tartrate (0.2 mg/kg body weight), and atropine sulfate (0.05 mg/kg body weight). Heparin (800 U/kg body weight) was administered intravenously a few minutes before cross-clamping the IVC for anticoagulation during anastomosis. The mean age of the animals upon the implantation of iTEV was 20.83 ± 1.069 months (range = 13.67–25.10 months).

Under general anesthesia, the seventh intercostal space was opened with a right thoracotomy incision. After heparinization, the IVC adjacent to the diaphragm and the right atrium adjacent to the right atrial–IVC junction were cross-clamped, and the IVC was incised and resected to anastomose the biodegradable scaffold to the greatest possible length using a simple clamping technique. The proximal end of the scaffold was sutured to the right atrium, and the distal end was sutured to the remaining IVC stump. Six excised native tissues were used for biochemical analysis.

The composite tubular scaffold was 10.5 mm in diameter, with a mean length of 26.03 ± 1.062 mm (23.00–33.00 mm). The tubular scaffold comprised multiple layers of L-lactide and ε-caprolactone copolymer (PLCL) sponge, spiraled fibers of 2–0 PLCL, and two 2–0 PLCL fiber warps placed in a long-axis direction for reinforcement to avoid constriction of the material after implantation. The outer side of the spiral was covered with 7–0 meshed PLCL fibers to support the PLCL sponge (Fig 1A–1C). Theoretically, the scaffold completely loses its

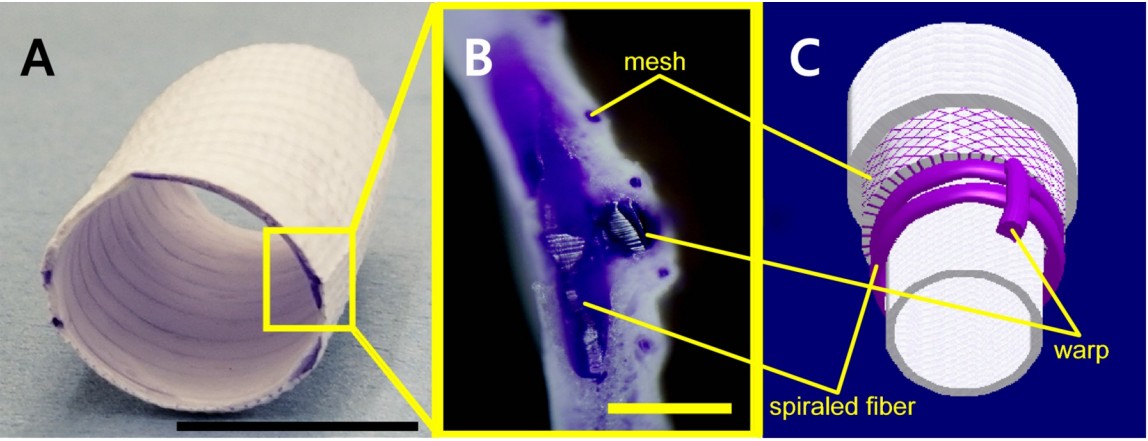

**Fig 1. Biodegradable scaffold and iTEV at 1 year post-implantation.** (A) Macroscopic images of the biodegradable scaffold used in this study (bar: 1 cm). (B) Higher magnification of a scaffold cross-section (bar: 500 μm). The spiraled fiber, the longitudinal warp, a cross-section of mesh fibers for scaffold reinforcement, and the honey-comb-shaped sponge are shown. (C) A schematic of the biodegradable scaffold in multiple layers is shown. From the inner side is the sponge composed of L-lactide and ε-caprolactone (PLCL) copolymer, a spiraled 2–0 PLCL fiber, two 2–0 PLCL fiber warps placed in a long-axis direction for reinforcement, a 7–0 meshed PLCL fiber, and the sponge.

strength within 6 months, and the material itself degrades within 8–12 months due to non-enzymatic hydrolysis [2, 5, 10].

## Angiography and pressure measurements

Ten animals underwent both pressure studies and angiography at 0.5, 1, 3, 6 (n = 10), and 12 (n = 8) months post-implantation. For the control studies, 10 untreated animals not related to the present study underwent pressure studies and angiography using the same protocol. Animals were anesthetized using the protocol described above and placed in sterilized conditions for femoral vein dissection. An 8-Fr long-catheter sheath (Terumo, Tokyo, Japan) was placed through the femoral vein to the IVC adjacent to the HV, as previously described [2, 5, 10]. Animals were then placed in a lateral position for angiography. VascuTape® Radiopaque Tape (LeMaitre Vascular GK, Tokyo, Japan) was used with centimeter-long reference markings on the chest. A mobile C-arm imaging system (WHA-200 OPESCOPE ACTIVO; Shimadzu, Tokyo, Japan) was used to identify the sheath and catheter locations and to perform digital angiography. Approximately 10–15 ml of angiographic agent was injected to evaluate the features of the iTEV, IVC, and HV with breath-holding to avoid respiratory fluctuation. No anti-coagulants were administered until euthanasia was performed [1–5, 10]. All angiographic data were saved to digital devices for further analysis.

Simultaneously, a Millar Mikro-Tip catheter (Millar Instruments Inc., Houston, TX) was used to continuously measure the pressure of the right atrium, the mid-portion of the iTEV, and the HV. Pressure waves were recorded using LabChart 8.1.5 data analysis software (ADInstruments Japan, Nagoya, Japan). The peak pressure of the right atrium, mid-portion of iTEV, and distal portion of iTEV were measured to analyze pressure gradients between the right atrium and the IVC. Ten samples of native IVC and HV from untreated (control) dogs were measured in the same manner as described above.

Digital angiographic imaging data were recorded in a lateral position. Data were used to measure the smallest internal diameter of iTEV and the angle formed by the distal sides of the IVC and HV (IVC–HV junction angle) using the Ruler tool in Photoshop CS6 Extended (Adobe, San Jose, CA). HV index, as a newly defined simple indicator for evaluating iTEV in a

time-course-dependent manner, reflected both morphometry and hemodynamic metrics of iTEV. HV index was calculated according to the following formula: HV index = (IVC–HV junction angle) ÷ ($\pi \times$ [minimal internal iTEV diameter ÷ 2]$^2$). To show the superiority of the HV index, the overall correlation between the pressure gradient and IVC–HV junction angle, the least estimated area of iTEV, and the HV index of iTEV were examined simultaneously.

## Macroscopic and histological examination

After careful dissection, macroscopic sample overview images were taken with a digital camera. The length of iTEV was measured directly between the two suture lines. iTEV was longitudinally dissected to perform histological and biochemical analyses.

For histological examination, samples were fixed in 4% paraformaldehyde at pH 7.0 in phosphate-buffered saline, embedded in paraffin, and sectioned to 4–5 μm. Some sections were subjected to hematoxylin and eosin, Masson's trichrome, Victoria blue van Gieson, or modified von Kossa stain. Immunostaining of remaining sections was performed using antibodies against factor VIII (1:2000; Abcam, Tokyo, Japan), α-SMA (clone 1A4, 1:2000; Dako, Tokyo, Japan), myosin heavy chain (MHC, 1:500; Sigma, St. Louis, MO), smooth muscle myosin heavy chain 1 (1:500; Yamasa, Tokyo, Japan), smooth muscle myosin heavy chain 2 (1:2000; Abcam), desmin (RD301, 1:500, ab8976; Abcam), vimentin (V9, 1:500, ab8069; Abcam), and MYH11 (1:200, ab53219; Abcam). Inflammatory cells in the specimens of iTEV at 1 and 12 months after implantation were identified by immunostaining for macrophage marker CD68 (1:500; Proteintech, Rosemont, IL). Histological examinations were performed as previously reported [1–5, 10].

All histological examinations and measurements were performed using light microscopy (Biozero Bz 8000; Keyence, Osaka, Japan), stereomicroscopy (MVX10; Olympus, Tokyo, Japan), and accompanying analytical software.

## Biochemical protein analysis

After thawing, iTEV and native IVC samples were weighed, and total protein was extracted using the Minute™ Total Protein Extraction Kit (Invent Biotechnologies Inc., Eden Prairie, MN). Total protein content was determined using the TaKaRa Bradford Protein Assay Kit (Takara, Kusatsu, Shiga, Japan). Equal amounts (10 μg) of denatured protein were added per lane for separation in 4%–15% Mini-PROTEAN® TGX™ Precast Gels (Bio-Rad, Tokyo, Japan) using Tris-glycine buffer (Bio-Rad) and transferred to polyvinylidene difluoride (PVDF) membranes (Trans-Blot® Turbo™ Mini PVDF Transfer Packs; Bio-Rad) using the Trans-Blot® Turbo™ Transfer System (Bio-Rad). Western BLoT Rapid Detect v2.0 (Takara) and antibodies were used to detect immobilized target proteins on PVDF membranes, according to the manufacturer's protocol. Sample immunoblots were incubated with antibodies against the following targets: α-SMA (1:1000, clone 1A4; Dako), MYH11 (1:1000, ab53219; Abcam), and desmin (1:1000, RD-301; ThermoFisher Scientific, Rockford, IL) or vimentin (1:2000, ab8069; Abcam) to detect three blots simultaneously on the same lane. Membranes were treated with a chemiluminescent reagent (Western BLoT Sensitive HRP Substrate; Takara) for enhancement, followed by analysis of blot images using a cooled charge-coupled device camera (Las 3000 Mini; Fujifilm, Tokyo, Japan) and image analysis software (Multi Gauge version 3.0; Fujifilm). MagicMark™ XP Western Protein Standard (Invitrogen, Carlsbad, CA) was used to determine protein size. Data were quantified as the ratio between the specified protein's luminescence and α-SMA expression.

To assess collagen content in iTEV, high-performance liquid chromatography was used to measure 4-hydroxyproline in samples, as previously described [2, 5, 10]. A commercially

available Fastin™ Elastin Assay kit (F2000; Biocolor, Antrim, UK) was used to quantify elastin in iTEV and native veins. Representative iTEV and native samples were weighed to obtain wet weight. Samples were then minced and digested at least three times with 0.25 M oxalic acid at 100°C for 1 h to extract mature insoluble elastin into the supernatant. Elastin content in each extract was measured using the Fastin Elastin Assay kit according to the manufacturer's instructions, and the results were normalized to sample wet weight to yield the mass of insoluble elastin per tissue wet weight (μg/mg). The results from 12-month iTEV samples and native veins (n = 8 each) were used in this study. Absorbance was determined at 513 nm using the SpectraMax ABS Plus microplate reader (Molecular Devices, Tokyo, Japan), and all samples were tested in duplicate in each assay.

## Statistical analysis

All data were tested for normality using the Shapiro–Wilk test ($P < 0.05$ indicates a non-normal distribution) and coefficients of skewness and kurtosis. The Wilcoxon rank-sum test was used to compare results between controls and 4-hydroxyproline or elastin samples. The Kruskal–Wallis test was used to evaluate statistical significance between samples at each time point. The Steel–Dwass test was used as a posthoc test to calculate pairwise comparisons of mean rank sums of independent samples between group levels. Spearman's rank correlation coefficient (denoted by ρ; Rho) was used to measure the strength and direction of the association between pressure gradient and the HV index, the least estimated area of iTEV, and the IVC–HV junction angle by time (months). Furthermore, the equivalency of independent correlation coefficients was determined to examine whether correlation coefficients between HV index and pressure gradients at each month differed.

The maximum information coefficient (MIC), the maximum asymmetry score (MAS), the minimum cell number, MIC-$R^2$, and the total information coefficient (TIC) were measured to test the linearity, monotonicity, complexity, and strength of the relationship between the pressure gradient and least estimated area of iTEV, the IVC–HV junction angle, and the HV index [11].

All data are expressed as mean ± standard error of the mean and/or median and range, and significance was assumed for differences with $P$ values of <0.05.

All data management and analyses were conducted using Microsoft Excel (Microsoft Japan Co., Ltd., Tokyo, Japan) and R 3.5.1 (R Core Team [2018]; R: A language and environment for statistical computing; R Foundation for Statistical Computing, Vienna, Austria).

## Results

### Macroscopic overview of iTEV

All animals examined in this study remained alive and healthy without any serious complications until euthanasia. No ascites due to narrowing or stenosis of iTEV was noted. All animals with implanted iTEV underwent angiography and pressure studies before dissection. Scaffolds, which were sewn between the right atrium and the IVC, remodeled into native-like vessels (Fig 4B). No thrombosis or aneurysmal changes in iTEV were observed at dissection. iTEV length at dissection, when compared with the implanted scaffold length, indicated that all iTEV grafts changed in length over a period of 12 months, with this change ranging from −1.3 mm to 2.0 mm (0.463 ± 0.018 mm; n = 8).

### Angiography and relationship between pressure gradient and HV index

All animals were placed in a lateral position for angiographic and hemodynamic studies. An example of chronological changes in iTEV over time is shown in Fig 2. As shown by the yellow

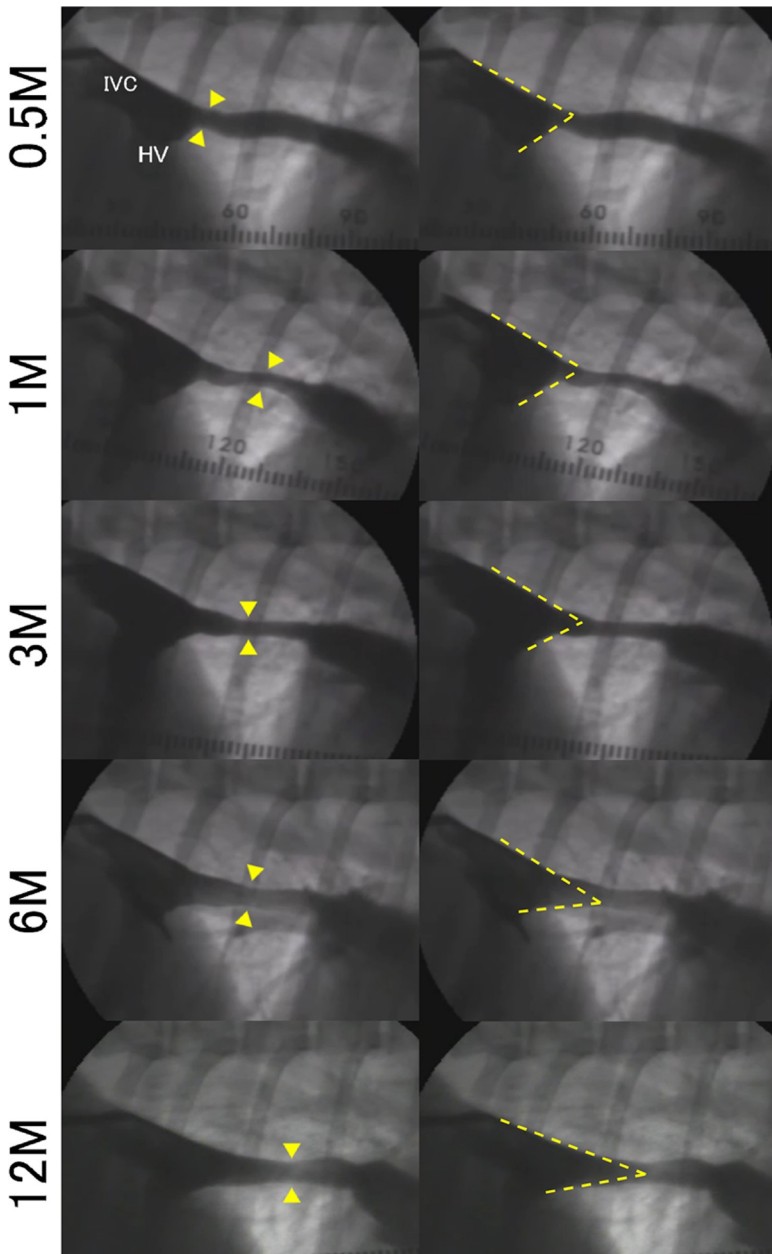

**Fig 2. Representative data of chronological changes in iTEV angiography in the same sample.** Left row: Yellow arrowheads indicate the site with the smallest iTEV diameter in each study. IVC: inferior vena cava, HV: hepatic vein. Right row: The angle was composed of two slanted dashed lines at the IVC–HV junction, which was measured to calculate the HV index. iTEV features, as well as native IVC, HV, and iTEV measurements change over time.

triangles and short dashed lines in Fig 2, all digital data were examined to calculate the smallest iTEV diameter, and the angle of the two dashed slanted lines was used to calculate HV index. Spearman's Rho of the least estimated area of iTEV, the IVC–HV junction angle, and HV index were −0.695 (95% confidence interval [CI]: −0.792, −0.540, $P < 0.001$) (S2A Fig), 0.603 (95% CI: 0.40, 0.757, $P < 0.001$) (S2E Fig), and 0.718 (95% CI: 0.560, 0.818, $P < 0.001$), respectively (Table 1 and Fig 3A). As shown in Table 2, HV index showed a strong correlation with

**Table 1. Difference in Spearman's Rho values by measuring methods.**

|  | Lower 95% CI | Upper 95% CI | Rho | *P* value |
|---|---|---|---|---|
| Angle | 0.405 | 0.757 | 0.603 | $5.41 \times e^{-07}$ |
| Area | -0.792 | -0.540 | -0.695 | $1.57 \times e^{-09}$ |
| HV index | 0.560 | 0.818 | 0.718 | $< 2.20 \times e^{-16}$ |

Spearman's Rho values of IVC–HV junction angle (Angle), least estimated area of iTEV (Area), and HV index. Lower and upper 95% confidence intervals (CIs) and *P* values are shown.

pressure gradient, monotonicity, linearity, the lowest MIC-$R^2$ value, and the highest TIC. Considering these results and the Rho value of the HV index, we adopted HV index in this study.

Next, we tested the correlation between HV index and pressure gradient by month. Spearman's Rho values of HV index of native IVC and iTEV at 0.5, 1, 3, 6, and 12 months were 0.370, 0.709, 0.539, 0.794, 0.600, and 0.786, respectively (Fig 3F). When the equivalency of independent correlation coefficients was calculated, there was no difference in Spearman's Rho value by month (Chi-squared value = 2.452, degrees of freedom = 5, *P* = 0.784). The pressure gradient of iTEV between the right atrium and the IVC is shown in Fig 3C. A significant difference in pressure gradient was only observed between native IVC (1.269 ± 0.236, median: 1.353, range: 0.022–2.162) and 1-month iTEV (4.693 ± 0.876, median: 4.656, range: 0.273–8.794) (*P* = 0.024). During tissue remodeling, although a change in scaffold diameter (S1 Fig and S1 Table) was observed, the scaffold maintained its patency with concurrent material degradation. A significant difference in HV index was observed between native IVC (0.297 ± 0.040, median: 0.301, range: 0.143–0.540) and 0.5-month iTEV (2.277 ± 0.436, median: 2.337, range: 0.281–4.543) (*P* = 0.003); 1-month iTEV (3.532 ± 0.589, median: 3.667, range: 0.752–6.532) (*P* = 0.0002); 3-month iTEV (3.381 ± 0.644, median: 3.241, range: 0.810–6.479) (*P* = 0.0002); 6-month iTEV (2.608 ± 0.515, median: 2.323, range: 0.599–5.056) (*P* = 0.0002); and 12-month iTEV (1.914 ± 0.541, median: 1.401, range: 0.644–5.126) (*P* < 0.001).

We measured HV index to evaluate time-dependent changes in each sample. Time-course changes in pressure gradient and HV index by sample are shown in Fig 3E and 3F. Individual differences are clearly shown because the peak pressure gradient appears between 1 and 6 months post-implantation. Furthermore, the pressure gradient of most samples decreased to native IVC levels; however, HV index recovery was not complete by 12 months post-implantation (Fig 3C to 3F).

Simultaneously, we tested the correlation between pressure gradient and the least estimated area of iTEV as well as the correlation between pressure gradient and the IVC–HV junction by month. Spearman's Rho values of the least estimated area of native IVC and iTEV at 0.5, 1, 3, 6, and 12 months were 0.188, –0.636, –0.261, –0.503, –0.770, and –0.905, respectively (S2B Fig). When the equivalency of independent correlation coefficients was calculated, there was no difference in Spearman's Rho value by month (Chi-squared value = 10.59, degrees of freedom = 5, *P* = 0.060). Spearman's Rho values of the IVC–HV junction angle of native IVC and iTEV at 0.5, 1, 3, 6, and 12 months were 0.285, 0.503, 0.903, 0.770, 0.455, and 0.571, respectively (S2F Fig). When the equivalency of independent correlation coefficients was calculated, there was no difference in Spearman's Rho value by month (Chi-squared value = 6.580, degrees of freedom = 5, *P* = 0.253).

The changes in least estimated area of iTEV and native IVC are shown in S2C Fig, and time-course changes in least estimated area by sample are shown in S2D Fig. Significant differences in the least estimated area were observed between native IVC (69.15 ± 7.014, median:

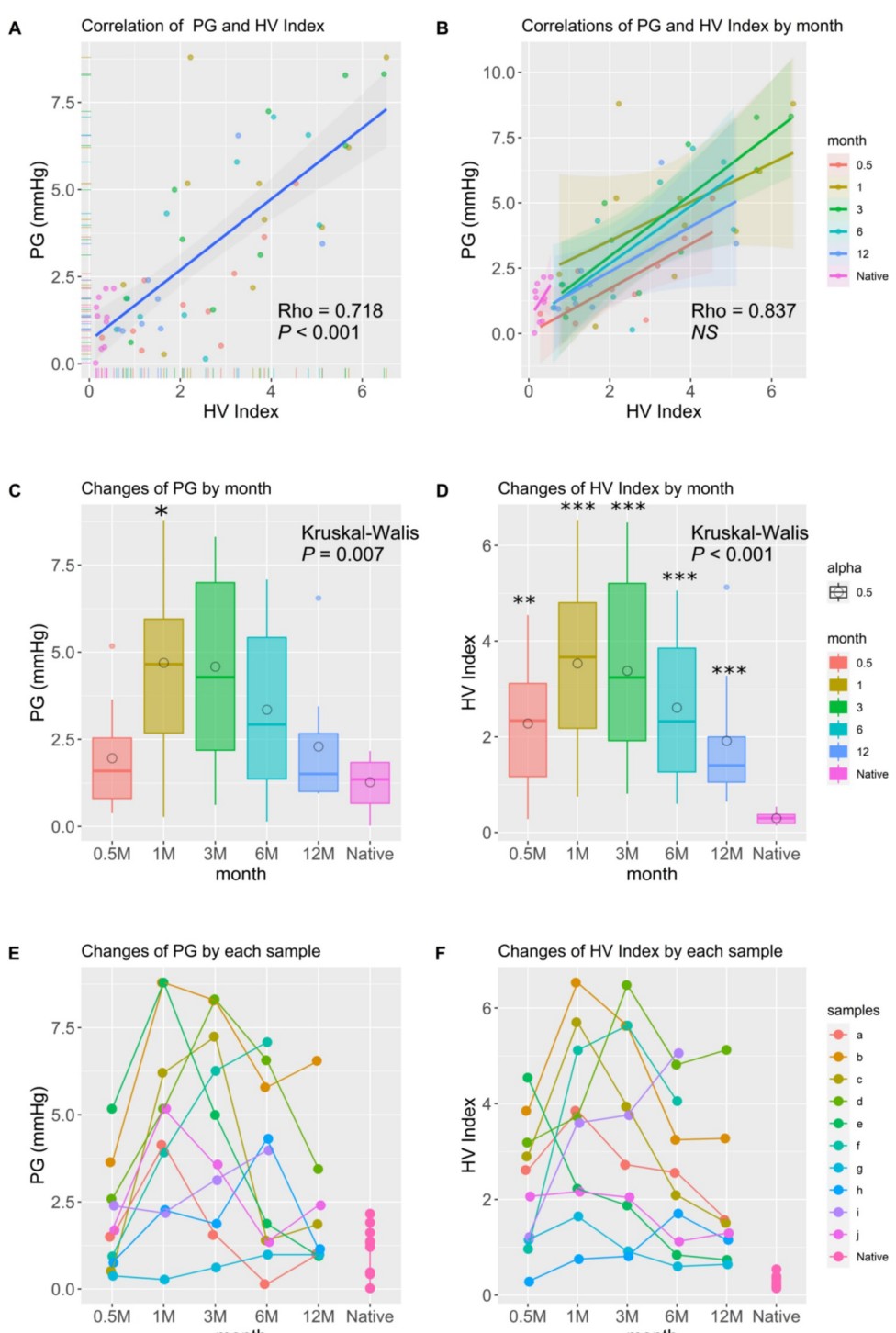

**Fig 3.** (A) Correlation between hepatic vein (HV) index and pressure gradient. Spearman's rank correlation Rho value of the HV index is shown. Rho = 0.718 (95% confidence interval: 0.564, 0.826; $P < 0.001$). (B) Correlation between HV index and pressure gradient by month. In native IVC, Rho = 0.370, and in iTEV at 0.5 months, Rho = 0.709; at 1 month, Rho = 0.540; at 3 months, Rho = 0.794; at 6 months, Rho = 0.600; and at 12 months, Rho = 0.786. There was no difference in the correlation between months ($P = 0.784$, NS: not significant). (C and D) Changes in pressure gradient (C) and HV index (D) by month expressed as box and whisker plots. Lines represent the lower, median, and upper quartile values. Whiskers represent the extent of the remaining data. Circles represent mean values. Pressure gradient, $P = 0.007$; HV index, $P < 0.001$ using the Kruskal–Wallis test. The Steel–Dwass test was used for posthoc analysis. (C) Native vs. 1 month (*, $P = 0.024$). (D) Native vs. 0.5 months (**, $P = 0.003$); native vs. 1 month (***, $P = 0.0002$); native

vs. 3 months (***, $P = 0.0002$); native vs. 6 months (***, $P = 0.0002$); and native vs. 12 months (***, $P = 0.0007$). (E and F) Each dot-line graph shows time-dependent changes in pressure gradient (E) and HV index (F) by sample.

63.48, range: 45.91–124.2) and 0.5-month iTEV (29.19 ± 3.695, median: 27.81, range: 15.77–55.60) ($P < 0.001$); 1-month iTEV (18.21 ± 2.667, median: 16.49, range: 9.700–37.59) ($P < 0.001$); 3-month iTEV (18.81 ± 2.149, median: 17.92, range: 11.97–28.89) ($P < 0.001$); 6-month iTEV (22.90 ± 3.158, median: 22.26, range: 11.69–40.67) ($P < 0.001$); and 12-month iTEV (28.30 ± 4.774, median: 24.89, range: 13.71–53.24) ($P = 0.002$) (S2C Fig). Changes in

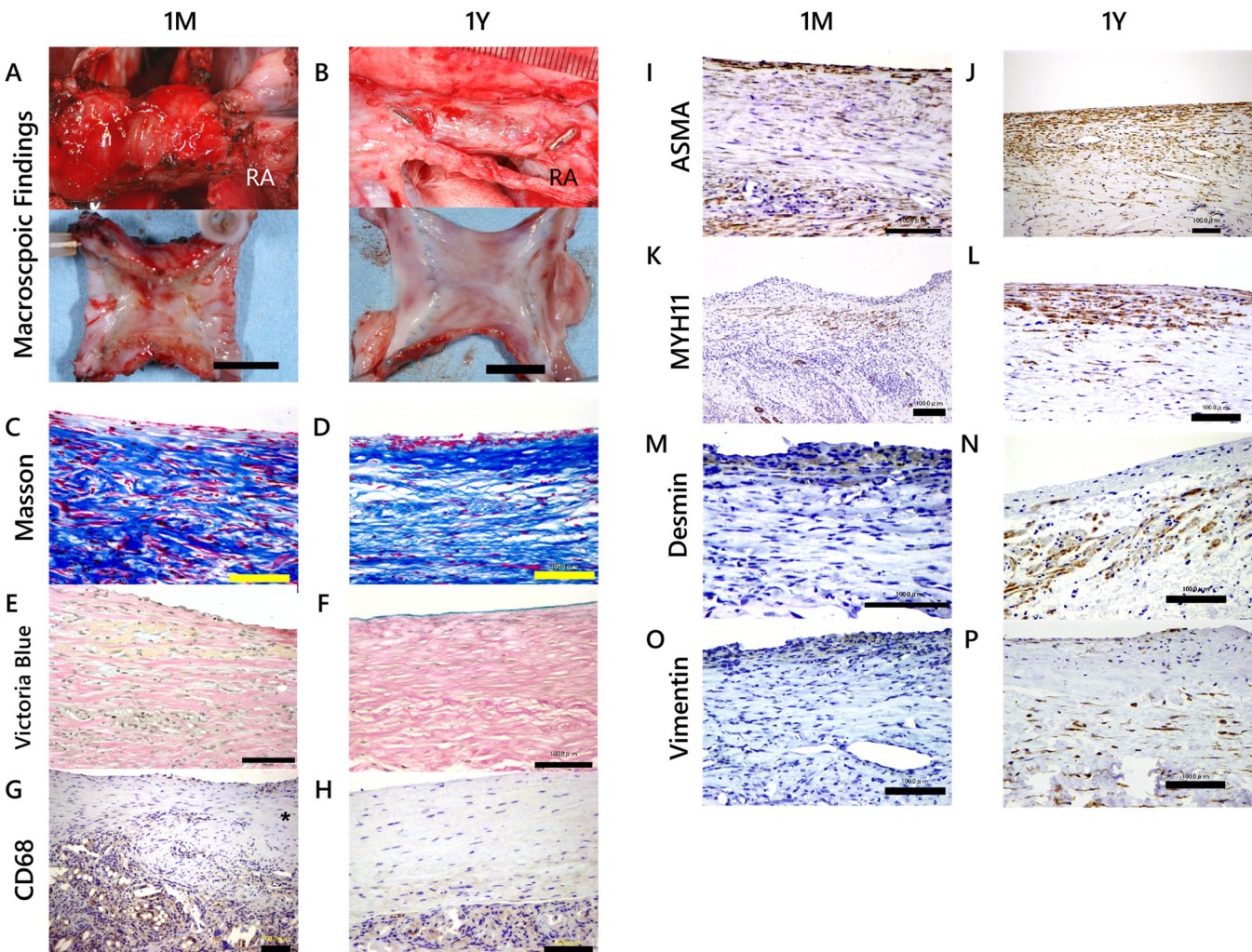

**Fig 4. Representative histological sections of iTEV at 1 month and 1 year post-implantation.** (A and B) Macroscopic view of 1-month iTEV (A) and 1-year iTEV (B). View of resected iTEV is shown in the lower panel (bars: 1 cm, RA: right atrium). (C and D) Masson's trichrome and (E and F) Victoria blue staining. (G and H) Inflammatory cells in the specimens of iTEV were identified by immunoexpression against CD68, a known macrophage marker, in developing iTEV (1 month) and mature iTEV (1 year) (bars: 100 μm). Histological studies revealed that CD68-positive cells were expressed mainly in the vicinity of the LCL fibers and sponge, and some were observed in the neo-intima that developed on the surface of the biodegradable scaffold () (1M; G). In 1-year iTEV, CD68-positive cells were observed on the outer side of the iTEV, which consisted of the remaining scaffold or adhesive tissues (1Y; H). (I and J) α-Smooth muscle actin (SMA), (K and L) myosin heavy chain 11 (MYH11), (M and N) desmin and (O and P) vimentin immunostaining demonstrate differences in expression in vascular smooth muscle cells (VSMCs) and intermediate filaments in developing iTEV (1 month) and mature iTEV (1 year) (bars: 100 μm). α-SMA expression is clearly visible at 1 month and 1 year, but MYH11, desmin, and vimentin are not clearly observed in 1-month iTEV. Mature VSMCs and intermediate filaments are more visible in 1-year iTEV.

**Table 2. The linearity, monotonicity, complexity, and strength of the relationship between the pressure gradient and three different measuring methods.**

|  | MIC | MAS | MCN | MIC.R$^2$ | TIC |
|---|---|---|---|---|---|
| Angle | 1 | 0.090 | 5.322 | 0.590 | 82.27 |
| Area | 1 | 0.198 | 5.000 | 0.711 | 91.58 |
| HV index | 1 | 0.097 | 4.585 | 0.425 | 95.86 |

The maximum information coefficient (MIC), the maximum asymmetry score (MAS), the minimum cell number (MCN), MIC-R$^2$, and the total information coefficient (TIC) of IVC–HV junction angle (Angle), least estimated area of iTEV (Area), and HV index are shown.

IVC–HV angle of iTEV and native IVC are shown in S2G Fig, and time-course changes in IVC–HV angle by sample are shown in S2H Fig. Significant differences in the IVC–HV angle were observed between native IVC (18.69 ± 2.861, median: 17.03, range: 9.130–41.54) and 0.5-month iTEV (54.09 ± 6.903, median: 58.74, range: 15.63–82.47) ($P = 0.013$); 1-month iTEV (54.75 ± 4.620, median: 53.45, range: 28.28–71.73) ($P < 0.001$); 3-month iTEV (52.70 ± 6.263, median: 53.06, range: 18.31–80.35) ($P = 0.003$); 6-month iTEV (47.04 ± 4.990, median: 51.80, range: 24.35–67.95) ($P = 0.002$); and 12-month iTEV (40.84 ± 5.005, median: 39.76, range: 26.86–70.26) ($P = 0.010$) (S2G Fig).

## Histological evaluation of iTEV

Representative macroscopic sections of iTEV at 1 month and 1 year post-implantation are shown in Fig 4A and 4B to compare SMC marker expression. Masson's trichrome staining (Fig 4C and 4D) shows SMCs in both 1-month and 1-year iTEV. Victoria blue staining (Fig 4E and 4F) shows elastin production in 1-year iTEV. In terms of inflammatory responses in iTEV, histological studies revealed that CD68-positive cells were expressed mainly in the vicinity of the LCL fibers and sponge one month after implantation. Some CD68-positive cells were observed at the developing intima that grew on the surface of the biodegradable scaffold (Fig 4G). As the fibers degrade with time, the number of CD68-positive cells also decreased significantly with time in the vessel wall of the iTEV; however, the CD68-positive cells were located on the outer side of the iTEV, which is composed of adhesive tissues and remaining fragments of biodegradable scaffold (Fig 4H).

α-SMA-positive cells are clearly shown in both 1-month and 1-year iTEV samples (Fig 4I and 4J). In contrast, MYH11 (Fig 4K), desmin (Fig 4M), and vimentin (Fig 4O) were expressed at low levels during the differentiation phase in 1-month iTEV samples. MYH11 (Fig 4L) and desmin (Fig 4N) exhibited higher expression levels in 1-year iTEV samples compared with 1-month iTEV samples.

## Increments of differentiated SMC markers to α-SMA protein

Representative immunoreactive band densitometry for MYH11 (227 kDa), desmin (54 kDa), vimentin (57 kDa), and α-SMA (40 kDa) in 1–12-month iTEV samples is shown in Fig 5A. Densitometry ratios of MYH11, desmin, and vimentin to α-SMA were measured at 1, 3, 6, and 12 months post-implantation. The Kruskal–Wallis test revealed changes in MYH11/α-SMA over time ($P = 0.025$), but changes in desmin/α-SMA ($P = 0.121$) and vimentin/α-SMA ($P = 0.228$) were not observed. The Steel–Dwass test revealed a difference in the densitometry ratio of MYH11/α-SMA between 1-month iTEV and native IVC ($P = 0.032$). A slight difference was also observed between 1-month and 12-month samples ($P = 0.052$).

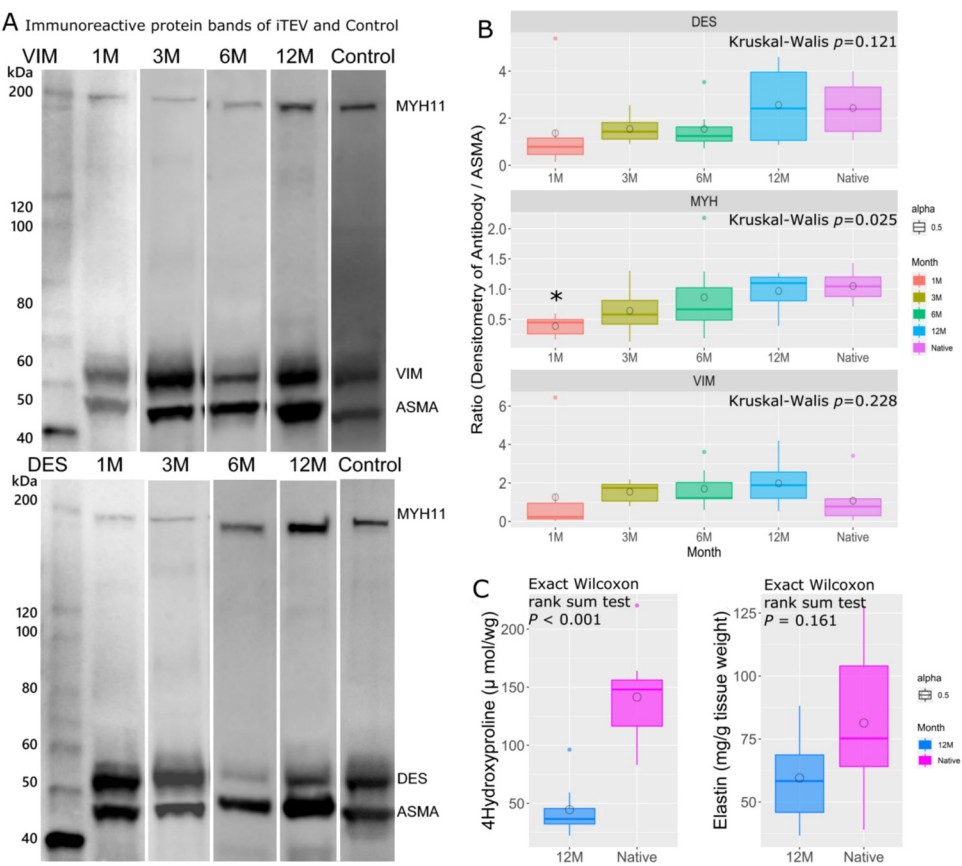

**Fig 5. Biochemical protein analysis of iTEV.** (A) Western blot immunoreactive protein bands for myosin heavy chain 11 (MYH11), vimentin, desmin, and α-smooth muscle actin (α-SMA) were measured by densitometry. Immunoreactive protein bands of MYH11 are visible in a 1-month iTEV sample. (B) Box and whisker plots of MYH11, vimentin, and desmin normalized to α-SMA in native IVC and iTEV samples by month. Lines represent lower, median, and upper quartile values. Whiskers represent the extent of the remaining data. Circles represent mean values. Desmin, $P = 0.121$; MYH11, $P = 0.026$; and vimentin, $P = 0.228$ using the Kruskal–Wallis test. The Steel–Dwass test was used for posthoc analysis. $^{*}$, $P < 0.05$ vs. native IVC. Box and whisker plots of 4-hydroxyproline (C) and elastin (D) in 12-month iTEV and native IVC samples. Lines represent lower, median, and upper quartile values. Whiskers represent the extent of the remaining data. Circles represent mean values. 4-Hydroxyproline, $P < 0.001$; elastin, $P = 0.161$ using the Wilcoxon rank-sum test.

## Biochemical findings

A significant difference in hydroxyproline content was found between 12-month iTEV and native IVC, with hydroxyproline content measuring 44.53 ± 8.310 μmol/g (median: 36.57 μmol/g) and 141.6 ± 15.38 μmol/g (median: 36.57 μmol/g) of wet tissue weight, respectively (Fig 5D; $P < 0.001$).

No difference in elastin content was observed between 12-month iTEV and native IVC, with elastin content measuring 59.55 ± 6.300 μg/g and 81.39 ± 10.78 μg/g of wet tissue weight, respectively (Fig 5E; $P = 0.161$).

## Discussion

After several modifications to the strategy of TEG procedures, we came to use a combination of a biodegradable scaffold and mononuclear bone marrow cells as a cell source to create TEGs in animal models and in the clinical setting of congenital heart surgery [3, 4, 7].

Acceptable results were obtained late after surgery [6–9], although material modifications were still required because not all patients achieved complete repair in terms of hemodynamics and graft features.

Postoperative evaluations depend on hemodynamic tests with invasive risks and morphometrics [12]. These evaluations are difficult because TEG features change dramatically with time as tissue and extracellular matrices develop to create mature vascular tissue. The diameter, size, and features of the original biodegradable scaffold are not maintained after implantation as the tissue develops, due to the unavoidable characteristics of tissue engineering, even though most TEGs spontaneously remodel into the size required by the living organism [12].

This time-dependent variation in TEG characteristics is due to differences in inflammatory responses [3] and individual growth potential because tissue regeneration mostly depends on the self-renewal ability of an organism. Moreover, grafts mostly decrease in diameter, and tissue development is strongly related to inflammatory reactions [3–5, 13]. Increased fibrous tissue and neotissue formation, followed by wall thickening, make it difficult to decide whether graft stenosis or narrowing has occurred.

We previously reported a scaffold that has less chance of exhibiting a decrease in graft diameter without any cell preparation to create TEGs with simplicity [2, 5, 10], which we named iTEV. iTEV demonstrated histological, biomechanical, and biochemical similarities to native vessels in a canine model with acceptable results. Chronological changes in iTEV observed through morphometric and pressure studies also differ between individuals. Even though iTEV eventually remodels into native-like tissue, inflammatory reactions followed by iTEV narrowing during tissue development are unavoidable, which might be a providence of nature.

An optimal evaluation method is thus needed to prove that TEGs grow and remodel with time, especially during the iTEV tissue generation period. Furthermore, it is difficult to evaluate TEG growth in clinical situations because the natural growth of children and the morphological/hemodynamic changes of TEGs do not always occur in parallel [6–9]. Moreover, because we must determine the need for further intervention, a simple but robust evaluation method during the tissue generation period is necessary, especially in a clinical setting. Cardiac echocardiography, though a simple and repeatable method, cannot accurately estimate the pressure gradient across TEGs used in extracardiac total cavo-pulmonary connection because the method is influenced by cardiac output and respiration, as well as several patient factors and technique-related factors. Considering these points, an appropriate evaluation method for graft function, which incorporates both hemodynamic and morphometric factors, is necessary for tissue-engineering vasculature in a venous position.

We focused on the specific characteristics of the iTEV tissue-generating process to develop an evaluation method. We also learned from our previous studies that changes in graft diameter and organismal growth do not always correlate [6–9]. Additionally, to develop a novel evaluation method for the evaluation of graft function, we had to consider the characteristics of iTEV development that differ between individuals. Consequently, we constructed a formula to calculate comparable data in each individual that are not influenced by the morphometrics and hemodynamics of iTEV, or by changes in body size. As shown in this study, HV index and pressure gradient are strongly associated; thus, we are convinced that HV index should be assessed as one of the evaluation methods during patient follow up after iTEV implantation.

We here demonstrate the usefulness of HV index in an animal model. We observed hemodynamic improvement, iTEV remodeling, and an increase in graft diameter simultaneously in this study. Tissue development differs between individuals, and HV index identified time-dependent changes in iTEV in individual samples. Moreover, HV index does not depend on the features of iTEV itself because it requires only the smallest graft diameter according to imaging data. Furthermore, the IVC–HV junction angle is not influenced by body size. Thus,

no matter how iTEV or the living body changes, previous and upcoming data of the HV index are comparable to evaluate changes in iTEV. With the smallest internal diameter and the IVC–HV junction angle, the HV index can be calculated using imaging data. This means that data obtained from different modalities, such as computed tomography and magnetic resonance imaging, could be comparable.

Novel TEG evaluation methods must be developed to achieve robust graft assessment and to demonstrate tissue generation and growth. A combination of routine pressure studies and measurement of objective data using non-invasive examinations to demonstrate TEG growth are desired, since it is difficult to explain TEG features by examining morphometry or hemodynamic metrics alone, especially in humans [12]. Changes in TEG size and patient growth, which influence hemodynamics, do not always occur in parallel [6, 8, 9]; thus, data do not always show TEG growth with good hemodynamic conditions. More specifically, size changes in the TEG do not always involve a reduction in the pressure gradient across the vasculature, and TEG function cannot be evaluated without hemodynamic metrics. However, systemic blood volume, pulmonary vascular resistance, cardiac function, sedation, and anesthetic agents also influence the pressure gradient during catheterization. Invasive hemodynamic monitoring is not a repeatable test because it poses several risks in children. Furthermore, repeat catheterization, which is invasive and sometimes has catastrophic adverse outcomes in children, should be avoided.

We set the standard limit of pressure gradient at 2 mmHg in a previous clinical trial of TEGs. However, this is not always appropriate because some patients require repeat intervention to minimize the pressure gradient across the vasculature during growth [6–9]. Furthermore, some patients developed unwanted veno-venous shunts, even with a low-pressure gradient across the TEG. Thus, we must decide whether changes in TEG diameter are due to growth and whether patients require further invasive postoperative intervention to recover graft stenotic changes, especially in a clinical setting. It is difficult to decide whether patients require intervention using classic pressure studies and the smallest TEG diameter value, due to aforementioned changes in TEG features and patient growth over time.

HV index demonstrated a good correlation with pressure gradient across the iTEV, and therefore changes in HV index may also be associated with an improvement/worsening in congestion in organs of the lower body, such as the liver, caused by iTEV narrowing or stenosis during tissue development. Furthermore, we show that HV index accurately reflects the in-vivo situation; specifically, HV index recovered more slowly compared with the pressure gradient.

Given the above, we present HV index as a novel measurement to evaluate iTEV (and TEGs) in terms of tissue development, growth, and remodeling. Additionally, the success of new vascular tissue formation will ultimately depend on the success of tissue regeneration and on the ability of the scaffold to meet the essential requirements of hierarchical composites of vascular tissue in each individual living organism. The increase in mature VSMC proliferation observed in this study confirmed that the results of the HV index are robust and objective, thereby demonstrating its usefulness for decision making in graft evaluation. We speculate that HV index would suggest not only graft growth and hemodynamic improvement, but also the efficacy of catheter intervention when introduced in a clinical setting. Moreover, we speculate that HV index could be applied to evaluate TEGs and for careful patient monitoring, such as detecting warning signs of graft narrowing or stenosis, evaluating pulmonary vascular resistance, and monitoring cardiac function, especially after the Fontan procedure in patients with congenital heart disease who undergo routine artificial conduit grafting.

## Study limitations

The present study is limited due to the small number of HV index measurements. Furthermore, angiographic data were measured only in a lateral position in this study. Therefore, measurements of IVC–HV junction angle in biplane studies might be preferable to calculate HV index for better evaluation of iTEV development.

## Conclusions

Tissue-engineered vasculature has specific dynamics after implantation, and therefore a novel evaluation method is desired for safe medical monitoring, especially in a clinical setting. HV index is a less invasive measurement modality compared with cardiac catheterization; it is not influenced by organismal growth and can easily identify improvements in TEGs by comparison with previous data. We believe that the HV index may yield great benefit postoperatively in children who require TEG implantation therapy.

## Supporting information

**S1 Fig. Minimal diameter of iTEV by each sample.** Time-course changes in minimal diameter of iTEV by sample are shown.
(TIF)

**S2 Fig.** (A) Correlation between the least estimated area of iTEV and pressure gradient. Spearman's rank correlation Rho value of the HV index is shown. Rho = -0.695 (95% confidence interval: -0.792, -0.540; $P < 0.001$). (B) Correlation between the least estimated area and pressure gradient by month (Estimated Rho = -0.542). In native IVC, Rho = 0.188, and in iTEV at 0.5 months, Rho = -0.636; at 1 month, Rho = -0.261; at 3 months, Rho = -0.503; at 6 months, Rho = -0.770; and at 12 months, Rho = -0.904. There was no difference in the correlation between months ($P = 0.060$, NS: not significant). (C) Changes in the least estimated area of iTEV by month expressed as box and whisker plots. Lines represent the lower, median, and upper quartile values. Whiskers represent the extent of the remaining data. Circles represent mean values. The least estimated area, $P < 0.001$ using the Kruskal–Wallis test. The Steel–Dwass test was used for posthoc analysis. Native vs. 0.5 month (*, $P < 0.001$); native vs. 1 month (***, $P < 0.001$); native vs. 3 months (***, $P < 0.001$); native vs. 6 months (***, $P < 0.001$); and native vs. 12 months (***, $P = 0.0016$). (D) Each dot-line graph shows time-dependent changes in the least estimate area by sample. (E) Correlation between the IVC-HV junction angle and pressure gradient. Spearman's rank correlation Rho value of the HV index is shown. Rho = 0.603 (95% confidence interval: 0.405, 0.757; $P < 0.001$). (F) Correlation between the IVC-HV junction angle and pressure gradient by month (Estimated Rho = 0.638). In native IVC, Rho = 0.285, and in iTEV at 0.5 months, Rho = 0.503; at 1 month, Rho = 0.903; at 3 months, Rho = 0.770; at 6 months, Rho = 0.455; and at 12 months, Rho = 0.571. There was no difference in the correlation between months ($P = 0.254$, NS: not significant). (G) Changes in the IVC-HV junction angle of iTEV by month expressed as box and whisker plots. Lines represent the lower, median, and upper quartile values. Whiskers represent the extent of the remaining data. Circles represent mean values. The IVC-HV junction angle, $P < 0.001$ using the Kruskal–Wallis test. The Steel–Dwass test was used for posthoc analysis. Native vs. 0.5 month (*, $P = 0.013$); native vs. 1 month (***, $P < 0.001$); native vs. 3 months (**, $P = 0.003$); native vs. 6 months (**, $P$ 0.002); and native vs. 12 months (*, $P = 0.010$). (H) Each dot-line graph shows time-dependent changes in the IVC-HV junction angle by sample.
(TIF)

**S1 Table. Raw data of minimal diameter of iTEV.** Raw data of time-course changes in minimal diameter of iTEV by dample are shown.
(TIF)

## Acknowledgments

We would like to acknowledge the efforts of Hisatoshi Saito and Toshio Watanabe for their animal surgeries and care, Kenji Yoshihara and Seiichi Kotoda for their skilled technical assistance in pressure studies, and Hiroaki Nagao for his contribution to the histological studies. We thank Emily Woodhouse, PhD, and Emma Longworth-Mills, PhD, from Edanz (https://jp. edanz.com/ac) for editing a draft of this manuscript.

## Author Contributions

**Conceptualization:** Goki Matsumura.

**Data curation:** Goki Matsumura, Noriko Isayama.

**Formal analysis:** Goki Matsumura.

**Funding acquisition:** Goki Matsumura.

**Investigation:** Goki Matsumura, Noriko Isayama.

**Methodology:** Goki Matsumura, Noriko Isayama, Hideki Sato.

**Project administration:** Goki Matsumura.

**Resources:** Goki Matsumura, Hideki Sato.

**Visualization:** Goki Matsumura, Noriko Isayama.

**Writing – original draft:** Goki Matsumura.

**Writing – review & editing:** Goki Matsumura, Noriko Isayama, Hideki Sato.

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
