## [Decision Letter · Decision Letter 0]

2 Feb 2022

PONE-D-21-26967Simple evaluation method for cell-free in-situ tissue-engineered vasculature monitoring: proof of growth and development in a canine modelPLOS ONE

Dear Dr. MATSUMURA,

Thank you for submitting your manuscript to PLOS ONE. After careful consideration, we feel that it has merit but does not fully meet PLOS ONE’s publication criteria as it currently stands. Therefore, we invite you to submit a revised version of the manuscript that addresses the points raised during the review process. Please submit your revised manuscript by Mar 19 2022 11:59PM. If you will need more time than this to complete your revisions, please reply to this message or contact the journal office at plosone@plos.org. Please include the following items when submitting your revised manuscript:A rebuttal letter that responds to each point raised by the academic editor and reviewer(s). You should upload this letter as a separate file labeled 'Response to Reviewers'.A marked-up copy of your manuscript that highlights changes made to the original version. You should upload this as a separate file labeled 'Revised Manuscript with Track Changes'.An unmarked version of your revised paper without tracked changes. You should upload this as a separate file labeled 'Manuscript'.

We look forward to receiving your revised manuscript.

Kind regards,

Atsushi Asakura, Ph.D

Academic Editor

PLOS ONE

https://journals.plos.org/plosone/s/file?id=ba62/PLOSOne_formatting_sample_title_authors_affiliations.pdf"

“This work was supported in part by KAKENHI (Grant-in-aid for Scientific Research) of the Japan Society for the Promotion of Science (JSPS) (C: number 17K10739, C: number 23591878) from The Ministry of Education, Culture, Sports, Science and Technology (MEXT), Japan, and in part by an Open Research Grant from the Japan Research Promotion Society for Cardiovascular Diseases (2018 and 2019). Any opinions, findings, conclusions, or recommendations expressed in this material are those of the author(s) and do not necessarily reflect the views of the author(s)’ organization(s), JSPS, or MEXT.” 

4. Thank you for stating the following in the Funding Section of your manuscript:

“This work was supported in part by KAKENHI (Grant-in-aid for Scientific Research) of the Japan Society for the Promotion of Science (JSPS) (C: number 17K10739, C: number 23591878) from The Ministry of Education, Culture, Sports, Science and Technology (MEXT), Japan, and in part by an Open Research Grant from the Japan Research Promotion Society for Cardiovascular Diseases (2018 and 2019). Any opinions, findings, conclusions, or recommendations expressed in this material are those of the author(s) and do not necessarily reflect the views of the author(s)’ organization(s), JSPS, or MEXT.”

“This work was supported in part by KAKENHI (Grant-in-aid for Scientific Research) of the Japan Society for the Promotion of Science (JSPS) (C: number 17K10739, C: number 23591878) from The Ministry of Education, Culture, Sports, Science and Technology (MEXT), Japan, and in part by an Open Research Grant from the Japan Research Promotion Society for Cardiovascular Diseases (2018 and 2019). Any opinions, findings, conclusions, or recommendations expressed in this material are those of the author(s) and do not necessarily reflect the views of the author(s)’ organization(s), JSPS, or MEXT.”

6. We note that you have indicated that data from this study are available upon request. PLOS only allows data to be available upon request if there are legal or ethical restrictions on sharing data publicly. For more information on unacceptable data access restrictions, please see http://journals.plos.org/plosone/s/data-availability#loc-unacceptable-data-access-restrictions.

7. Please amend your list of authors on the manuscript to ensure that each author is linked to an affiliation. Authors’ affiliations should reflect the institution where the work was done (if authors moved subsequently, you can also list the new affiliation stating “current affiliation:….” as necessary).

Reviewers' comments:

Reviewer's Responses to Questions

**Comments to the Author**

1. Is the manuscript technically sound, and do the data support the conclusions?

Reviewer #1: No

2. Has the statistical analysis been performed appropriately and rigorously? 

Reviewer #1: I Don't Know

3. Have the authors made all data underlying the findings in their manuscript fully available?

Reviewer #1: No

4. Is the manuscript presented in an intelligible fashion and written in standard English?

Reviewer #1: Yes

5. Review Comments to the Author

Reviewer #1: This manuscript reported the new evaluation method of tissue engineered blood vessels in the IVC position in dog model. Twenty-six animals underwent IVC replacement surgeries with in-site tissue-engineered vasculature (iTEV). These data are valuable and important for all the scientists working on vascular tissue engineering. However, there are several questions and concerns in this manuscript.

Their data is not supporting their conclusion because control data is missing. The title is also misleading and not appropriate from their data (it is complicated, not simple and nobody would use this index as this is only applicable in dog model). In addition, the reviewer do not think that this HV index is useful for evaluation of iTEV. The HV angle would be changed by the timing of respiration, table rotation angle at angiography. Even individual anatomical variety would be also related. The reviewer also could not understand the reason why they plan to utilize their own Index, because simplest evaluation method would be the measurement of vascular diameter and length over time. This manuscript also need additional information in control animal group without spiral reinforcement to prove their concept.

I have suggestions and questions to improve this manuscript.

1. Describe the narrowest diameter of tubular grafts after implantation over time., in all cases.

2. Describe the age of animals at the implantation of iTEV.

3. Describe the changes in body weight of animals at 12 months.

4. Describe the reason why the authors used many different size grafts from 23-33 mm in length, although animal body weight is very similar. They should have used the graft with same diameter and same length.

5. HV Index means the angle divided by narrowest area. Describe the reason why the authors did not check the co-relation between the angle and PG, or between narrowest area and PG.

6. Show the angiography and HV Index of control animal (Gore-Tex tube interposition) without any stenosis in Figure 2.

7. If the authors believe this HV Index is useful to evaluate the stenosis of the grafts, they can make a simulation banding model in control animals instead of graft implantation. Please comment on this issue.

8. When revising this manuscript, the reviewer would recommend that the authors should forget and remove this index part, just focus on the remodeling process of iTEV in histology and mechanical properties over time.

9. Describe the reason why the authors chose spiraled shape for reinforcement instead of ringed grafts.

10. Explain the reason why the fiber should be colored in blue.

11. In the abstract and Figure 4, the authors followed the animal at 0.5, 1, 3, 6, and 12 months, but in the Materials and Methods part (page 5), there are no mentioning about the group of 0.5 month. Did the authors survive animals after catheterization and angiography at 0.5 month after implantation? Or The authors made additional group?

12. In figure 4, histology should be shown with macroscopic pictures. 1month’s data looked like more stenotic from the data in Figure 3. Inflammatory cells like macrophage should be stained at 1 month and 1 year.

13. When the authors compare the histology and PG at 1 month and 3 months after implantation, is there any relations between the inflammation & thickening of tissue and PG?

6. PLOS authors have the option to publish the peer review history of their article (what does this mean?). If published, this will include your full peer review and any attached files.

Reviewer #1: No

---

## [Author Response · Author response to Decision Letter 0]

17 Mar 2022

Response to the Reviewer: 

Reviewers' comments:

Reviewer #1: This manuscript reported the new evaluation method of tissue engineered blood vessels in the IVC position in dog model. Twenty-six animals underwent IVC replacement surgeries with in-site tissue-engineered vasculature (iTEV). These data are valuable and important for all the scientists working on vascular tissue engineering. However, there are several questions and concerns in this manuscript.

Their data is not supporting their conclusion because control data is missing. The title is also misleading and not appropriate from their data (it is complicated, not simple and nobody would use this index as this is only applicable in dog model). In addition, the reviewer do not think that this HV index is useful for evaluation of iTEV. The HV angle would be changed by the timing of respiration, table rotation angle at angiography. Even individual anatomical variety would be also related. The reviewer also could not understand the reason why they plan to utilize their own Index, because simplest evaluation method would be the measurement of vascular diameter and length over time. This manuscript also need additional information in control animal group without spiral reinforcement to prove their concept.

I have suggestions and questions to improve this manuscript.

1. Describe the narrowest diameter of tubular grafts after implantation over time, in all cases.

We appreciate your suggestion. To describe the narrowest diameter of iTEV after implantation over time, we have constructed a figure and table that include these data. Please see S1: Fig 1, Minimal diameter of iTEV and S2: Table 1, Numerical raw data of minimal diameter of iTEV.

2. Describe the age of animals at the implantation of iTEV.

In accordance with your suggestion, we have added information regarding the age of the animals at the implantation of iTEV. Please see page 7, line number 149–151 of the revised manuscript.

3. Describe the changes in body weight of animals at 12 months.

We apologize that we do not have accurate data regarding this point. However, we measured the body weight of each animal upon every catheter examination to determine the adequate dose of the anesthetic agents. Thus, we note that the body weight at 12 months did not change markedly when compared with the body weight upon the initial surgery.

4. Describe the reason why the authors used many different size grafts from 23-33 mm in length, although animal body weight is very similar. They should have used the graft with same diameter and same length.

We used the same diameter of the scaffold for each implantation; however, in terms of length, we placed the vascular clamp as distal as possible to implant a longer scaffold. Because of anatomical variations, the length of the implanted scaffold differed.

5. HV Index means the angle divided by the narrowest area. Describe the reason why the authors did not check the co-relation between the angle and PG, or between narrowest area and PG.

To this point, we have added Tables 1 and 2 to demonstrate that the HV index yielded the best results to speculate the PG of iTEV. In accordance with your suggestion, we have also included data regarding the correlation between the angle and PG, and between the narrowest area and PG calculated with Spearman’s rank correlation. Please see S3 Fig.

6. Show the angiography and HV Index of control animal (Gore-Tex tube interposition) without any stenosis in Figure 2.

Thank you for your suggestion. We have revised Figure 2 to include the HV index of the control animal.

7. If the authors believe this HV Index is useful to evaluate the stenosis of the grafts, they can make a simulation banding model in control animals instead of graft implantation. Please comment on this issue.

It is valuable to know the relationship between the minimal diameter of iTEV and PG; however, in an acute experiment, the angle of HV will not change. We believe that the relationship between banded diameter and PG can be tested in a mock circuit, and therefore unnecessary animal experiments can be avoided. In terms of experiments that involve banding the IVC of the experimental animals, though the response is dependent on the tightness of the banding, we believe that banding the IVC results in rapid stress to the liver of the animal. If it is possible to control the banding after the operation through such means as adjustable banding, and if we are able to obtain adequate data with intermittently repeated catheter studies, then this approach may serve as a good control. However, considering animal welfare, and the principle goal of lessening the number of experiments performed on animals that involve the sacrifice of life, it might not be a useful experiment. Conversely, iTEV decreases its internal diameter, but the changes are gradual and followed by improvement. Accordingly, the experimental animals are able to endure the biological responses to the reduction of iTEV diameter over time. 

8. When revising this manuscript, the reviewer would recommend that the authors should forget and remove this index part, just focus on the remodeling process of iTEV in histology and mechanical properties over time.

The concept of this study is to explore a method that is not influenced by the morphological/hemodynamical changes of iTEV and/or size of the living body to examine small changes in iTEV over time through imaging data and demonstrate iTEV improvement. Thus, we evaluated the correlation between the pressure gradient (PG) of iTEV and several measurements obtained by imaging data to speculate PG differences in the time course. As the reviewer has mentioned, the HV index seems to be applicable only in the dog model. However, we believe that the HV index can also be applied in patients who have undergone total cavo-pulmonary connection (TCPC: Fontan type operation). According to our previous data in the clinical trials, when images from catheter study, 3D-image computed tomography, and/or magnetic resonance imaging are acquired from the front side, we can calculate the HV index to compare the present and previous data to evaluate the improvement/worsening of hemodynamics. Additionally, there is a chance to avoid repeated catheter studies in the patient with this method. Furthermore, in terms of laboratory experiments, histological developments can be speculated without the sacrifice of the experimental animals. In terms of manufacturing the new scaffold, this method can be applied to assess the quality of improved scaffolds in this IVC model, with less sacrifice of experimental animals. Because the changes in vascular diameter are so small, as shown in the present study, it is difficult to determine the remodeling process without catheter pressure study; nonetheless, catheter pressure study and angiography are affected by factors including respiration, the direction of irradiation, and changes in physical status. Thus, we concluded that the HV index is useful, but because several pitfalls of catheter pressure study remain, we performed angiography upon breath-holding to avoid respiratory fluctuation. 

Furthermore, when the biodegradable scaffold is applied in a patient who requires TCPC, there are no data to indicate the suitable size of the graft in growing patients with a size-changing tissue-engineered graft. Thus, a post-operative monitoring method to evaluate the condition of the tissue-engineered graft is desirable. There are multiple factors that influence the pressure gradient of the graft in Fontan patients, including pulmonary resistance, left side AV valve regurgitation, LV end-diastolic pressure, LV function, and collaterals. With a graft diameter and/or PG, total graft function is not fully evaluated. Thus, we performed this study to explore a new evaluation method. For these reasons, we would like to include the HV index in this manuscript.

9. Describe the reason why the authors chose spiraled shape for reinforcement instead of ringed grafts.

In our experimental design, we initially attempted to create a biodegradable scaffold that imitated a ringed Gore-Tex graft. It is possible to create ring-shaped biodegradable material; however, it is impossible to achieve similar mechanical strength with biodegrading ring-shaped materials. Furthermore, it was difficult to determine the optimal ring-to-ring distance. Ultimately, it was difficult to use the ring-shaped structure in our manufacturing process. Because we have studied vascular tissue engineering with spiraled biodegradable scaffolds since 2005 using the same implantation methods and protocol, we elected to use the fiber created with p(LA/CL) with marketing approval in this study. 

10. Explain the reason why the fiber should be colored in blue.

As mentioned above, we used fiber from commercially available surgical sutures as a raw material. There is no reason for the specific color of the fiber. However, the use of colored fiber is advantageous because it facilitates easier recognition of the fiber’s location and helps prevent pricking of the rigid fiber when suturing the graft. 

11. In the abstract and Figure 4, the authors followed the animal at 0.5, 1, 3, 6, and 12 months, but in the Materials and Methods part (page 5), there are no mentioning about the group of 0.5 month. Did the authors survive animals after catheterization and angiography at 0.5 month after implantation? Or The authors made additional group?

As mentioned in page 8 line 169-, 10 dogs underwent catheterization study at 0.5, 1, 3, 6, and 12 months. We did not include 0.5-month iTEV samples in histological and biochemical studies because our previous studies showed that two weeks post-implantation was an insufficient amount of time to observe histological/biochemical changes in the iTEV. Furthermore, excluding this time point allowed us to minimize the number of the experimental animals to be sacrificed. 

12. In figure 4, histology should be shown with macroscopic pictures. 1month’s data looked like more stenotic from the data in Figure 3. Inflammatory cells like macrophage should be stained at 1 month and 1 year.

In accordance with your helpful advice, we have revised Figure 4 to include macrophage evaluation at 1 month and 1 year. Thank you for your suggestion.

13. When the authors compare the histology and PG at 1 month and 3 months after implantation, is there any relations between the inflammation & thickening of tissue and PG?

Thank you for this insightful question. As shown in Figure 3 and S1 Fig, the time-dependent changes in pressure gradient, minimal diameter of iTEV, HV index, etc., differ in each sample and have no regularity in terms of tissue regeneration, especially within 6 months after implantation. This might be related to the differences in inflammatory response in each individual sample. According to our previous study, the wall thickness of iTEV seems strongly related to the internal diameter; thus we used the minimal area of iTEV to calculate the HV index. Conversely, we might speculate the wall thickness of iTEV from the internal diameter. However, the relationship between PG and wall thickness of iTEV was not evaluated in this study because we did not sacrifice animals at 1 and 3 months for the HV index study; therefore, some of the opinions written here are only speculative.

---

## [Editor Report · Decision Letter 1]

6 Apr 2022

Evaluation method for cell-free in-situ tissue-engineered vasculature monitoring: proof of growth and development in a canine IVC model

PONE-D-21-26967R1

Dear Dr. MATSUMURA,

We’re pleased to inform you that your manuscript has been judged scientifically suitable for publication and will be formally accepted for publication once it meets all outstanding technical requirements.

Kind regards,

Atsushi Asakura, Ph.D

Academic Editor

PLOS ONE
---

## [Editor Report · Acceptance letter]

8 Apr 2022

PONE-D-21-26967R1 

Evaluation method for cell-free in situ tissue-engineered vasculature monitoring: proof of growth and development in a canine IVC model 

Dear Dr. MATSUMURA:

I'm pleased to inform you that your manuscript has been deemed suitable for publication in PLOS ONE. Congratulations! Your manuscript is now with our production department. 

Kind regards, 

on behalf of

Dr. Atsushi Asakura 

Academic Editor

PLOS ONE